# The Interplay between Cell-Extracellular Matrix Interaction and Mitochondria Dynamics in Cancer

**DOI:** 10.3390/cancers14061433

**Published:** 2022-03-10

**Authors:** Bian Yanes, Elena Rainero

**Affiliations:** School of Biosciences, The University of Sheffield, Western Bank, Sheffield S10 2TN, UK; b.yanes@sheffield.ac.uk

**Keywords:** mitochondria dynamics, tumor microenvironment, extracellular matrix

## Abstract

**Simple Summary:**

Mitochondria are essential cellular organelles, involved in controlling energy production, cell metabolism, cell growth, and cell death. Since cellular functions are de-regulated in cancer, it is not surprising that mitochondria dysfunctions have been observed in this disease. One key aspect in controlling tumor formation and progression is the interaction between the cancer cells and their surrounding environment, known as the tumor microenvironment. The extracellular matrix is an abundant component of the tumor microenvironment and it has been shown to affect tumor initiation and progression. Here, we will explore how the interaction between cancer cells and the extracellular matrix impinges on mitochondria function during cancer progression.

**Abstract:**

The tumor microenvironment, in particular the extracellular matrix (ECM), plays a pivotal role in controlling tumor initiation and progression. In particular, the interaction between cancer cells and the ECM promotes cancer cell growth and invasion, leading to the formation of distant metastasis. Alterations in cancer cell metabolism is a key hallmark of cancer, which is often associated with alterations in mitochondrial dynamics. Recent research highlighted that, changes in mitochondrial dynamics are associated with cancer migration and metastasis—these has been extensively reviewed elsewhere. However, less is known about the interplay between the extracellular matrix and mitochondria functions. In this review, we will highlight how ECM remodeling associated with tumorigenesis contribute to the regulation of mitochondrial function, ultimately promoting cancer cell metabolic plasticity, able to fuel cancer invasion and metastasis.

## 1. Introduction

Cancer cells can change their metabolism according to their progression and their needs which can be provided by the interaction with the surrounding microenvironment [1]. Even though glycolysis has been found to be increased in many types of cancer, the role of mitochondria in cancer metabolism and progression has recently started to be of interest. Mitochondria are essential organelles which play multiple roles in various cell functions such as cell proliferation, apoptosis, and cell metabolism. They also regulate intracellular calcium homeostasis beside their major role which is generating ATP [2]. Due to their multiple functions, mitochondrial dysfunctions can play a role in various human diseases including cancer [2].

Mitochondria are highly dynamic organelles where their structure, function, and cellular distribution can reflect the metabolic changes. Their dynamics are usually represented by two processes, fission and fusion. Fission is a process where mitochondria are divided into multiple structures to generate new ones, thereby facilitating mitochondria relocation to cellular regions of high energy demand. Fission is mainly regulated by Dynamin-related protein (Drp1). Fusion is a process where numerus mitochondria are fused including damaged ones and is regulated by mitofusins (Mfns) and optic atrophy-1 (OPA1). When fusion reduces, the fragmentation of mitochondria into small, spherical mitochondria results in impaired respiratory function, while stimulation of fusion results in the formation of a highly connected mitochondria network [2,3,4].

The dysregulation in mitochondria fission and fusion processes affects both mitochondria structure and function, and it has been linked with human disease. Indeed, mutations in the regulators of mitochondria dynamics have been linked with several neurodevelopmental and neurodegenerative diseases. Mitophagy, which is a type of autophagy-promoting damaged mitochondria elimination, plays a role in the quality control and morphology of mitochondria [5,6]. In cancer, the dysregulation in mitochondria dynamics has been found to play a role in tumorigenesis. For instance, fragmented mitochondria have been observed in many types of cancer including breast cancer, melanoma, and pancreatic cancer as a result of an increase in DRP1 expression. Interestingly, this has been shown to be required for cancer cells migration. In addition, when cancer stem cells (CSC) rely on mitochondria respiration to generate ATP, they change mitochondria structure from elongated to fragmented. This has been found to play a role in CSC differentiation. On the other hand, quiescent CSCs keep their mitochondria in a fused structure [7].

Tumors are surrounded by multiple components which form the tumor microenvironment (TME). The TME consists of different cell types such as endothelial cells, immune cells, and fibroblasts, in addition to the extracellular matrix (ECM) which provides mechanical and biochemical support to the resident cells [8]. ECM plays a critical role in multiple cell functions such as cell proliferation, cell migration, and invasion. During tumorigenesis, the ECM plays both a tumor suppressive and a tumor promoting role. During the initial phases, the ECM is thought to mainly represent a barrier to cancer cell growth and migration. However, it has now been established that, as tumorigenesis progresses, the dysregulation of ECM dynamics and composition promotes tumor development [8]. The interaction between the tumor cells and the ECM is a fundamental factor that enhances tumorigenicity. Thus, understanding the crosstalk between tumors and the surrounding ECM is an essential criterion in understanding cancer progression.

Due to the high tumor growth rate and the limited blood supply, the tumor microenvironment has been found to be hypoxic and deprived of nutrients [9]. Therefore, cancer cells can reprogram their metabolism and adopt nutrient scavenging strategies to survive. It has already been shown that the ECM plays a role in cancer metabolism where cancer cells can uptake ECM proteins, degrade them into the lysosomes and use them as source of nutrients. Furthermore, the forces generated from cell-ECM interaction can regulate nutrient signaling, glucose, and lipid metabolism [10,11,12]. In addition, the availability of extracellular nutrients can also affect mitochondrial dynamics. It has been previously shown that tumor cells grown in Hanks’ Balanced Salt Solution (HBSS, a low glucose medium) tend to keep their mitochondria in a connected elongated structure. Mechanistically, this was mediated by protein kinase A (PKA)-mediated DRP1 phosphorylation at Ser637, resulting in DRP1 inactivation. As a consequence, glucose unavailability promoted a switch in cancer cell metabolism from glycolysis to the mitochondrial oxidative phosphorylation (OXPHOS), facilitating cell survival [13].

Accumulating evidence suggest that there is a link between the ECM and mitochondria and that mitochondria can sense any changes in the TME including changes in the ECM such as its composition and stiffness. In this review, we will discuss the interplay between the ECM and mitochondria and its role in controlling mitochondria morphology and dynamics.

## 2. Adhesion Signaling Modulates Mitochondria Dynamics and Function

ECM forms most of the TME in many tumors and it comprises hundreds of proteins—altogether defined as the “matrisome” manly secreted by cancer cells themselves and by the cancer-associated fibroblasts (CAFs) [14]. The matrisome is composed of more than 1000 genes, encoding for ECM and ECM-associated proteins. The “core matrisome” is composed of ~300 protein, while the remaining proteins are ECM-modifying enzymes, ECM-binding growth factors, and other proteins that are found to be associated with the ECM. These proteins interact and bind with each other to form the three-dimensional structure of the ECM. There are two types of the ECM; the basement membrane (BM) mainly composed of collagen IV, laminins, and nidogen which separates the epithelium from the stroma, and the interstitial matrix, mostly composed of collagen I, proteoglycans, and fibronectin which is considered as the tissue structural scaffolding [8]. 

Cancer cells interact with the ECM via cell membrane receptors such as integrins. Integrins are heterodimers which consist of two subunits, α and β. They are activated by binding to their ECM ligands which then trigger multiple signaling pathways that promote cell proliferation and survival [15]. Integrins link the ECM with the actin cytoskeleton via adhesion complexes, known as integrin adhesion complexes (IACs) [10]. Several mass-spectrometry based approaches have characterized the composition of the “adhesome”, composed of over 24,000 proteins, 60 of them defined as core proteins. When cells adhere to the substrate, the binding of integrins to ECM components results in the recruitment of core adhesion proteins, which include talin, vinculin, focal adhesion kinase (FAK), paxillin, integrin-linked kinase (ILK), PINCH, parvin, kindlin, α-actinin, zyxin, and vasodilator-stimulated phosphoprotein (VASP). Following the formation of initial focal complexes, these mature to mechanically regulated focal adhesions. The centripetal movement of focal adhesion along fibronectin fibers results in the formation of fibrillar adhesion, which are mainly enriched in tensins. Altogether, they bridge the ECM with the actin cytoskeleton and play pivotal roles in controlling cell adhesion, migration, proliferation, and survival [15,16,17].

More recently, it has been established that adhesion complexes also play an important role in controlling nutrient sensing. In particular, the mammalian target of rapamycin (mTOR) complex 1 (mTORC1) has been shown to be activated at paxillin-containing focal adhesion, through the upregulation of localized amino acid uptake. In addition, fibronectin-bound integrin has been shown to be internalized specifically from fibrillar adhesion in a tensin-dependent manner. This internalization pathway supports lysosomal clustering and mTOR activation in ovarian cancer cells. Finally, the energy sensor AMPK has been shown to prevent tensin-dependent integrin activation in fibroblasts [10,18,19]. Interestingly, adhesion complexes have also been found to play a role in controlling mitochondria function.

Paxillin (PAX) is a focal adhesion protein that interacts with FAK, promoting FAK-Src binding and the downstream signaling pathways to link integrins to the actin cytoskeleton [17]. It has been shown that PAX can be mutated, amplified, or overexpressed in lung cancer and the effect of the most common PAX mutants on mitochondrial dynamics have been tested. The mutants (A127T, P233L, and P487L) showed an effect on focal adhesions while the mutants (P233L and D399N) showed an association with the anti-apoptotic protein B cell lymphoma2 (BCL-2), which is known to localize to the mitochondria, and with DRP1 and MFN2 (Figure 1A). Some of the PAX mutations also caused changes in the mitochondrial dynamics; the A127T and the P487L mutants triggered dense mitochondria with a network structure while P52L, and the D399N mutants caused a mitochondria fragmentation (Figure 1A) [20]. However, these mutants’ effect was tested in vitro utilizing the human embryonic kidney epithelial cells HEK-239. It would be interesting to see if this is applicable to different sets of lung cancer cell lines and to other cancer types that feature mutated PAX including breast, and colorectal cancers. It is still not known how these changes in mitochondria morphology induced by PAX mutations impinge on mitochondria function and energy production [21]. In addition, phosphoinositide 3-kinase (PI3K) inhibition has been shown to promote the targeting of mitochondria to phosphorylated FAK, resulting in increased assembly and turnover of focal adhesions. This has been shown to be regulated by Akt, mTOR, and oxidative phosphorylation [22]; however, the detailed molecular mechanisms await to be elucidated.

PINCH1 and 2 are focal adhesion proteins expressed in mammalian cells [23]. They interact with ILK and Parvin to form the PINCH-ILK-Parvin (PIP) complex that links integrins to the actin cytoskeleton [23,24]. PINCH1 has been found to be overexpressed in lung cancer [24]. Knockout of PINCH1 in lung adenocarcinoma cells resulted in increased DRP1 expression and therefore mitochondria fission, which led to a reduction in the level of pyrroline-5-carboxylate reductase 1 (PYCR1), an essential enzyme in proline synthesis, therefore reducing proline synthesis (Figure 1B) [24]. PINCH1 effect on mitochondria might be indirect by affecting Kindlin-2 translocation from the cytosol into mitochondria. Kindlin-2 is known to localize to focal adhesions and stiff ECM can trigger its translocation to the mitochondria where it can interact with PYCR1, increasing proline synthesis, collagen matrix formation, and cell proliferation (Figure 1B) [24,25]. However, knockout of Kindlin-2 did not affect DRP1 level, but it increased mitochondria fission [24]. Therefore, there might be other mechanisms which can affect mitochondria dynamics through Kindlin-2. More studies are needed to identify the molecular mechanisms that regulate Kindlin-2-dependent mitochondria fission. PINCH has also been found to be highly expressed in brains of patients with neuroinflammatory diseases and a previous study by Natarajaseenivasan et al. showed that inflammation can increase PINCH expression, which in turn affects its binding with Parvin and disrupts the formation of PIP complex. Therefore, Parvin binds to the actin-associated kinase testicular protein kinase 1 (TASK1) causing its deactivation, which in turn reduces the phosphorylation of cofilin leading to actin depolymerization. This actin depolymerization can cause mitochondria mis-localization mediated by the disruption of the kinesin-Trak-Miro complex that connects mitochondria to the actin cytoskeleton [26]. Hence, PINCH can not only affect mitochondria dynamics but also its localization to the actin cytoskeleton. 

In cultured endothelial cells, αvβ3 integrin has also been shown to support mitochondria function, via the activation of a FAK-signal transducer and activator of transcription 3 (STAT3) signaling pathway induced by vitronectin binding [27]. STAT3 has been found to be phosphorylated at Ser727 by FAK in the cytoplasm then translocated to the mitochondria, where it interacts with electron transport chain (ETC) complexes I, II, and V. This interaction has been shown to prevent the loss of mitochondrial membrane potential, therefore maintaining mitochondria function (Figure 1C). Inhibiting FAK reduced ATP production, the reserve capacity and the respiratory capacity, hence completely abolished mitochondria function, suggesting that adhesion signaling can regulate mitochondria function [27]. However, the pathways downstream FAK still needs to be identified and it is still unknown how this influences any of the cell functions or metabolism.

Integrin-mediated cell adhesion has been linked with reactive oxygen species (ROS) production. ROS are chemicals that can be generated from different intracellular sources including mitochondria (mtROS) and they contain the hydrogen radical (OH), the superoxide anion (O_2_^−^), and the hydrogen peroxide (H_2_O_2_) [28]. The level of ROS is known to be elevated in cancer and plays a role in cancer development [2]. ROS-induced signaling pathways result in cell damage and cell death. However, cancer cells can circumvent this by upregulating the expression of antioxidant proteins. In addition, low levels of ROS can regulate metabolic pathways and cell proliferation [2,28]. Interestingly, an integrin-mediated signaling pathway has been identified by Werner and Werb that can induce ROS production. In Rabbit fibroblasts, changes in the cell shape due to blocking α5β1 integrin function using an anti-α5 mAb can activate Rac or RhoA GTPases which in turn indirectly increase the production of ROS from mitochondria (Figure 1D) and promote the production of both collagenase 1 (CL1) and the activation of nuclear factor k B (NFkB) [29]. Interestingly, the production of ROS did not cause apoptosis induction, but it decreased the expression of proinflammatory cytokines via NFkB activation [29]. Future studies are needed to investigate if these Rho GTPase-mediated mitochondria changes could also happen in cancer as changes in cancer cells shape are triggered during tumor progression. 

Mitochondria can also influence the ability of integrins to bind to their ligands; indeed, the dysfunction of the mitochondrial OXPHOS can increase the glycosylation of α5β1 integrin, promoting the binding to its ligand fibronectin and therefore enhancing the cells metastatic potential [30]. 

## 3. The Cytoskeleton Modulates Mitochondria Function

One of the factors that can affect metabolism is the extrinsic and the intrinsic forces that can be generated either from cell-ECM contact, cell–cell contact, or shear stresses [31]. These forces can generate signaling pathways that regulate multiple cell functions such as cell shape, cell proliferation, and migration [12,17]. The cell cytoskeleton plays a critical role in translating the ECM-generated forces into signals. The actin filaments, microtubules, and the intermediate filaments are the main components of the cytoskeleton and are usually organized into networks [32]. However, they can be reorganized as a response to forces such as the ones that can be generated by ECM remodeling, which has been found to be increased in cancer [32]. ECM remodeling can affect the biophysical and the biochemical properties of ECM, including its stiffness due to collagen crosslinking mediated by the lysyl oxidase (LOX) enzyme. The increase in ECM stiffness in turn can generate extrinsic forces and activate integrins. This results in focal adhesion maturation, coupled with actin polymerization. This process is mainly promoted by the recruitment of paxillin and vinculin to focal adhesions and results in the formation of dense actin stress fibers, which in turn can induce traction forces on ECM [33,34]. 

The response to the forces not only plays a role in controlling cell behavior but can also regulate the structure and the function of cell organelles such as mitochondria [12]. Several studies showed how the cytoskeleton can participate in mitochondria function and dynamics. DRP1 localization at the mitochondria is necessary for the fission event in U2OS osteosarcoma and HeLa cervical cancer cells. DRP1 recruitment is mediated by the actin cytoskeleton as it has been previously shown that the endoplasmic reticulum-bound formin protein (INF2), which plays a role in the actin polymerization and depolymerization, polymerizes actin after interacting with Spire1C, an actin nucleator protein. The intrinsic forces that are generated from this interaction increase the recruitment of myosin II which initiates mitochondria constriction and facilitates DRP1 binding, therefore promoting the mitochondria fission event (Figure 2A) [35,36,37,38].

Another mechanism by which actin participate in mitochondria fission was shown by Li et al. in HeLa cells. In a DRP1-dependent fission event, F-actin accumulates transiently on the outer mitochondrial membrane (OMM) and this accumulation is controlled by the actin-modifying proteins cofilin, cortactin, and actin related protein 2/3 (Arp2/3) complexes (Figure 2B) [39]. This was followed by a study using breast cancer cells showing that cofilin, which is overexpressed in many types of cancer, is not only necessary for the initiation of mitochondria fission but also for the regulation of mitophagy, a form of autophagy which results in the selective degradation of damaged mitochondria. Indeed, cofilin localization to the mitochondria has been shown to decrease the mitochondria membrane potential and trigger mitophagy mediated by PTEN-induced kinase 1 (PINK1)/parkin RBR E3 ubiquitin protein ligase (PARK2) pathway (Figure 2C) [40]. It would be interesting to study whether this has any effect on breast cancer cells proliferation and/or migration. In addition, as this study was performed in the absence of ECM, more studies are needed to investigate the effect of ECM stiffness on cofilin-dependent mitochondria fission.

## 4. The Effect of ECM/Mitochondria Interplay on Cell Migration

### 4.1. The Role of Mitochondria Trafficking in Cancer Metastasis

Metastasis is a multi-step process where cancer cells migrate from the primary site, invade into the ECM, enter the blood vessels, and reach a secondary site [15]. Cancer cells can migrate either individually or as a group and both ways require the involvement of ECM, integrin adhesion proteins, and the cytoskeleton [15]. The formation of lamellipodia is one of the key steps in cell migration and high energy production is required for the assembly of actin filaments at the leading edge of the cell [3]. Microtubule-and actin-based transport motors are used by cells to cluster their mitochondria at the leading edge, allowing for the localized generation of ATP to power cell migration. 

The dysregulation in mitochondrial dynamics has been linked to cancer invasion. Indeed, mitochondria fission and the expression of Drp1 have been found to be enhanced in highly invasive breast cancer cells. The fission event is necessary for the redistribution and the movement of mitochondria to the lamellipodia region of the cells, where the energy demand is high. The formation of lamellipodia was opposed by inhibiting DRP1, as well as cell migration and the invasion after performing matrigel invasion and migration assays [3]. These results indicate that mitochondrial dynamics play a role in cancer cells migration, however, as matrigel is a basement membrane extract [41], it would be interesting to use other types of ECM to investigate if mitochondria trafficking is regulated in an ECM-dependent manner. Similarly, in ovarian cancer cells it has been shown that mitochondria are trafficked to the leading edge of the cells on microtubules to fulfil the ATP demand that the migrating cells need to infiltrate into a 3D matrix and this trafficking was shown to be regulated by the metabolic sensor AMP-activated kinase (AMPK). Indeed, AMPK was activated in 3D invasive protrusions by the rapid ATP utilization at this location and this was required for mitochondria trafficking during 3D cell invasion [42]. Interestingly, several AMPK targets have been implicated in organelle trafficking and microtubule transport, and AMPK has been involved in the control of mitochondria fission [43,44]. Further work is required to characterize the molecular mechanisms through which localized AMPK activation drive mitochondria trafficking. 

Mitochondria trafficking to the leading edge of the cells has also been found to be a way for cancer cells to bypass PI3K inhibitors and this was accompanied by a growth-factor receptor-mediated Akt and mTOR re-phosphorylation. The accumulation of energetically active elongated mitochondria at the leading edge promoted membrane dynamics, focal adhesion turnover, and cell migration. Interestingly, the inhibition of the mitochondria fusion effect on mitofusion1 opposed mitochondria trafficking and cells invasion induced by PI3K inhibitors [22]. This suggests that, depending on the context, both mitochondria fission and fusion can control the organelle accumulation at the leading edge and cell invasion. 

### 4.2. The Effect of ECM Stiffness on Mitochondria

The role of mitochondria in cancer cells migration and metastasis has been extensively reviewed elsewhere [45,46]. However, one of the main factors that can affect cancer cell migration is ECM stiffness. The stiffness of ECM depends on its composition and the organization of its components, and these two factors can change according to the tumor type and its location. The mechanical and the physical properties of ECM can be affected by its stiffness which in turn can affect tumor invasion and migration [15]. The stiffness of ECM has been found to be increased in several types of cancer including breast cancer. Moreover, Morris et al. showed that the composition and the stiffness of collagen I ECM can reprogram breast cancer metabolism. Indeed, the highly metastatic breast cancer 4T1 cells demonstrated an increase in the expression of tricarboxylic acid (TCA) cycle genes leading to enhanced glutamine metabolism and use glutamine to fuel the TCA cycle in presence of high-density compared to a low-density collagen I (Figure 3A). By contrast there was a decrease in the glycolysis-mediating genes [47].

The ECM architecture can also affect the ATP:ADP ratio inside the cells. By increasing collagen I density from 2D to 3D matrices and by changing it from aligned to random fibers, the ATP:ADP ratio increased inside the cells (Figure 3B) [48]. This increase in ATP:ADP ratio is suggested to be due to an increase in OXPHOS activity to fulfil the energy demand for cytoskeletal remodeling needed for cell migration in 3D matrices [49]. It should be noted that in the previous studies only one simple type of ECM was used (collagen I). As the physiological ECM is a 3D complex matrix, it would be interesting to assess whether similar results can be obtained by using more complex 3D matrices. 

The stiffness of ECM has been found to play a role in mitochondria dynamics by two different mechanisms; soft collagen I has been shown to trigger mitochondria fission in human mesenchymal stem cells (MSCs) in a kindlin-2 dependent process, while stiff ECM can trigger mitochondria fusion and an inhibition of DRP1 in a PINCH1-dependent manner (Figure 3C) [50]. More studies are needed to characterize which molecules, downstream of PINCH1 and Kindlin-2, are involved in ECM-dependent mitochondria dynamics and how this can affect cell behavior. 

During pancreatic cancer progression, ECM deposition is dramatically increased, and ECM stiffness in this context has been shown to affect mitochondria dynamics. In ductal adenocarcinoma cell lines, mitochondria become more elongated in stiff ECM, and they accumulate in invasive protrusions to fulfil the ATP demand associated with cytoskeletal remodeling and cell migration [51]. ATP recycling is used by cells to maintain local ATP gradients and is mediated by the phosphocreatine (pCr)–creatine kinase (CK) system, where creatine can be phosphorylated to phosphocreatine, an energy storage molecule, which can transfer the phosphate to ADP to regenerate ATP. This reaction is catalyzed by the cytoplasmic creatine kinase B-type (CKB) which expression has been found to be increased by the mechanical cues generated from stiff ECM, in an integrin and YAP signaling-dependent manner (Figure 3D) [51]. A recent study showed that in cancer, stiff ECM can activate a cell stress response mediated by heat shoch factor1 (HSF1), which usually regulates the cell response to heat stress. The physical stress generated from stiff ECM triggers an integrin-Rho-associated protein kinase (ROCK) mechanosignaling pathway resulting in the activation of SLC9A1 (Solute carrier family 9 member A1), which regulates the cytosolic pH and pH-dependent downstream signaling, leading to ROS induction. However, the cells can overcome the elevated ROS by increasing HSF1 activity and expression which in turn triggers changes in mitochondria structure, composition, and function. Indeed, HSF1 enhances mitochondria fragmentation, mitochondria potential, and the expression of mitochondria import machinery [52]. Thus, cancer cells can overcome the stress generated from stiff ECM through changes in mitochondria structure and function. 

Several anti-cancer therapies target cell proliferation and growth, however metabolic reprograming and plasticity of cancer cells enable them to restore their ability to grow and survive. Thus, understanding the tumor metabolic adaptabilities will be beneficial to finding new targets which could improve the existing therapies and avoid the potential resistance. Interestingly, ECM stiffness has been found to play a role in cancer cells resistance to chemotherapy. A recent study by Romani et al. showed that, being in a soft microenvironment, metastatic breast cancer cells can overcome oxidative stress and ROS-dependent chemotherapy drugs via the transcription factor NRF2 (nuclear factor erthroid-2-related factor 2). Soft ECM enhances mitochondria fission due to increased expression of DRP1 and mitochondria elongation factor 1 and 2 (MIEF1/2). In this case, mitochondria fission increases both the production of mtROS and the activity of NRF2 antioxidant transcriptional response, therefore cells increase their tolerance to oxidative stress [53]. It will be interesting to investigate the signaling pathways that are involved in this process.

## 5. The Crosstalk between the ECM and Mitochondria Function

One of the main aspects that the cells need to control is the balance between survival and apoptosis. ECM detachment results in cell apoptosis through a process called anoikis [54]. Mitochondria play a key role in apoptosis as the permeabilization of the outer mitochondrial membrane can release multiple proteins such as cytochrome c which can activate caspases in the cytosol leading to cell death [55]. In pancreatic cancer cells, mitochondria function was altered after ECM detachment, and this was due to mitochondria depolarization and the release of proapoptotic molecules which in turn caused necrosis (another type of cell death) (Figure 4A). The mitochondrial dysfunction was inhibited after plating pancreatic cancer cells on laminin or fibronectin, as these ECM proteins increased the mitochondria membrane potential and inhibited the release of the proapoptotic molecules (cytochrome c and Smac/DIABLO). Therefore, pancreatic cancer cells can survive by attaching to ECM proteins [56].

It has been found that ECM-detached cells can activate the receptor-interacting protein kinase 1 (RIPK1), which in turn induces mitophagy, mediated by the mitochondrial phosphatase phosphoglycerate mutase 5 (PGAM5), leading to ROS induction and a non-apoptotic cell death (Figure 4B) [57]. However, according to the human protein atlas, the expression of RIPK1 is high in most types of cancer. Hence, either cancer cells are able to find a way to escape the non-apoptotic cell death triggered by RIPK1 or mitophagy can promote cancer cells survival, as through mitophagy damaged mitochondria can be eliminated [57].

In myopathic mice, a deficiency in collagen VI changed mitochondria structure causing a mitochondrial dysfunction mediated by abnormal opening of the permeability transition pore (PTP) and skeletal muscles apoptosis [58,59]; interestingly, it has been suggested PTP desensitization could rescue muscle atrophy [60]. The molecular mechanism that regulates this effect was not identified; however, it was suggested that this mitochondria dysfunction might be caused by the disruption in collagen VI-integrin interaction, as a same effect has been identified in fibroblasts. In fibroblasts, adding soluble collagen VI to the media had an anti-apoptotic effect mediated by β1 integrin. This was accompanied by a down regulation of Bax (a pro apoptotic protein) and an upregulation of cyclin A, B, and D1 which can enhance the progression through the cell cycle [61]. This anti-apoptotic effect of collagen VI might also help cancer cells to survive. This is particularly relevant for tumors in which the expression of collagen VI is high, such as in melanoma. Knockdown of fibronectin has also been found to affect mitochondria function. It has been shown by Wu et al. that siRNA- and shRNA-mediated silencing of fibronectin induced mesangial cells (MC) apoptosis. This was mediated by the mitochondria, and it could be at least in part due to intracellular Ca^2+^ alterations. However, what is the role of Ca^2+^ in this process and how it is regulated still need to be investigated. By adding exogenous fibronectin, the apoptotic process was reversed [62]. It is well established that high levels of Ca^2+^ can trigger apoptosis via the opening of the mitochondria permeability pores and the release of cytochrome c [2]. Therefore, it would be interesting to investigate what is the role of Ca^2+^ in fibronectin-induced MC apoptosis and how it is regulated. Cardiac myocytes are embedded in the ECM. During cardiac development and disease, ECM remodeling has been found to influence mitochondria function. This was observed by an increase in ATP production, and mitochondria basal respiration when the ECM rigidity was increased [63,64].

Interestingly, changes in mitochondria functioning can also affect ECM remodeling. It has been shown by Waveren et al. that ECM remodeling factors such as the metalloproteases enzymes MMPs and their inhibitors tissue inhibitors of proteases (TIMPs) can be regulated by OXPHOS. In this study, in an OXPHOS-deficient osteosarcoma cells, an increase in MMP1 and a decrease in TIMP1 were observed, suggesting a bi-directional interaction between mitochondria and ECM (Figure 4C) [65]. However, how this is regulated and what the molecular mechanism is, still need to be investigated. In breast cancer, it has been found that the expression of the mitochondria transmembrane receptor TMEM126A is downregulated, resulting in mitochondria depolarization and an increase in ROS production. Furthermore, TMEM126A is also suggested to play a role in ECM remodeling as its downregulation is accompanied by an alteration in cell adhesion and ECM protein genes (Figure 4D) [66]. Interestingly, its downregulation also promoted breast cancer cells migration, through the induction of actin cytoskeleton rearrangement. The upstream signaling regulation was not identified; however, it is suggested that the expression of TMEM126A might be regulated by p53 or FOXP3 [66]. This supports the idea of a bi-directional and reciprocal regulation between mitochondria dysfunction and cell-ECM adhesion.

## 6. Conclusions

Altered metabolism has been identified as one of the hallmarks of cancer [67] where cancer cells have different metabolic behaviors than normal cells. This allows cancer cells to better adapt to the challenging TME, optimizing the way they obtain the energy required for cell proliferation and migration. The role of mitochondria in cancer has recently started to be of interest especially in cancer metastasis, where mitochondria trafficking to the leading edge of the cells is required to fulfil the ATP demand associated with cell invasion. The TME and the ECM play a pivotal role in cancer cells metabolism. Cell-ECM adhesion has been shown to control nutrient signaling and the endocytosis and degradation of ECM components has been recently established as a scavenging strategy for the acquisition of nutrients [10]. In this review, we highlighted the interaction between the ECM and the mitochondria and how ECM stiffness, structure, and the forces that are generated from ECM can affect mitochondria function and dynamics. In particular, we have highlighted a bi-directional interplay between mitochondria and the ECM. On the one hand, cell-ECM interaction plays and important role in controlling mitochondria functions, either directly or through the modulation of the actin cytoskeleton. This is particularly relevant in the metastatic process, where mitochondria represent the main energy source at the leading edge to fuel the cell migration machinery. On the other hand, OXPHOS and mitochondria depolarization have shown to impact the ECM, through the modulation of MMP-dependent ECM degradation and cell-ECM adhesion. However, this is an emerging field and further studies are needed to identify the role of 3D in vivo-like ECM on mitochondria and energy production. The signaling pathways involved in this crosstalk could pave the way for novel therapeutic strategies aimed at limiting cancer growth and metastasis.

## Figures and Tables

**Figure 1 cancers-14-01433-f001:**
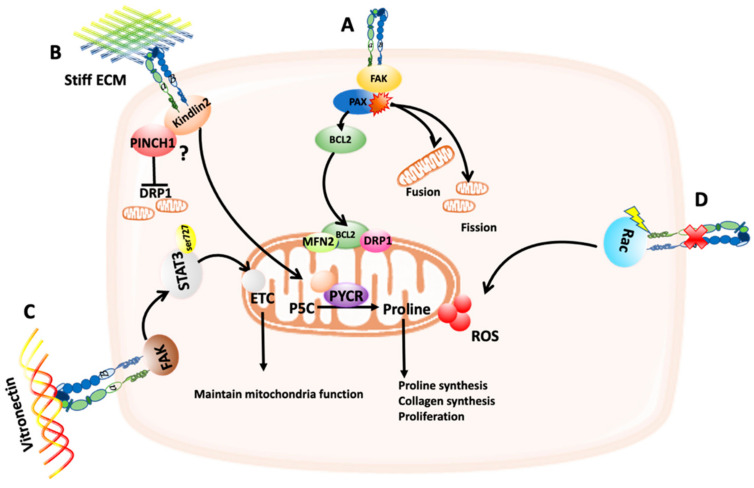
Schematic representation of the effect of cell/extracellular matrix (ECM) adhesion on mitochondria function and dynamics. (**A**) Mutated forms of paxillin (PAX) have been shown to associate with BCL2, which in turn localizes to the mitochondria and interact with DRP1 and MFN2. In addition, mutations in PAX can cause changes in mitochondria fission and fusion. (**B**) PINCH1/kindlin2 interaction can affect DRP1 expression, therefore preventing mitochondria fission. Stiff ECM can trigger the translocation of kindlin2 to the mitochondria, where it interacts with PYCR1, increasing proline synthesis, collagen synthesis, and cell proliferation. (**C**) Binding of αvβ3 integrin to its ligand vitronectin can trigger FAK-dependent phosphorylation of STAT3 at Ser727, which promotes STAT3 translocation to the mitochondria, where it interacts with ETC complexes, maintaining mitochondria function. (**D**) Blocking α5β1 integrin results in Rac activation, which cause an increase in ROS induction.

**Figure 2 cancers-14-01433-f002:**
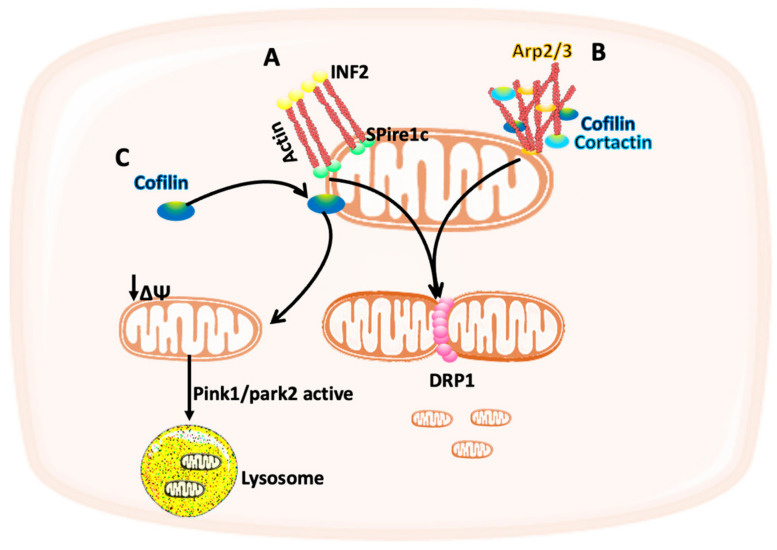
Schematic representation of the interplay between mitochondria and the cytoskeleton. (**A**) The intrinsic forces generated from actin polymerization upon INF2 and spire1 binding can facilitate DRP1 recruitment to the mitochondria, therefore promoting mitochondria fission. (**B**) Cofilin, cortactin, and actin related protein 2/3 (Arp2/3) complexes regulate actin filaments accumulation on the outer mitochondrial membrane, which can initiate mitochondria fission. (**C**) Cofilin localization at the mitochondria can reduce the mitochondria membrane potential and trigger mitophagy mediated by Pink1/park2.

**Figure 3 cancers-14-01433-f003:**
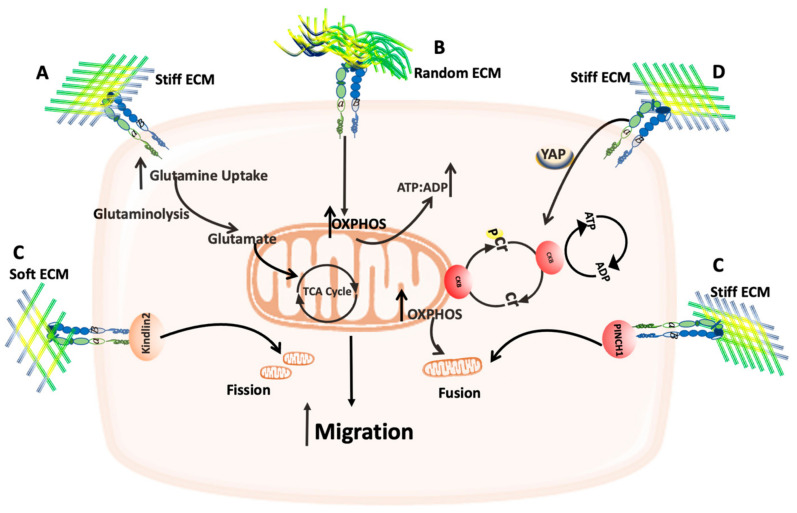
Schematic representation of the effect of ECM/Mitochondria interplay on cell migration. (**A**) Stiff ECM can enhance glutamine uptake, which in turn feeds the TCA cycle. (**B**) Random 3D ECM can increase OXPHOS activity and the ATP:ADP ratio. (**C**) Soft ECM can trigger mitochondria fission in a Kindlin2-dependent manner, while stiff ECM can trigger mitochondria fusion in a PINCH1-dependent manner. (**D**) Stiff ECM can increase CKB expression and activity in a YAP signaling-dependent fashion, which in turn increases the OXPHOS activity and mitochondria fusion.

**Figure 4 cancers-14-01433-f004:**
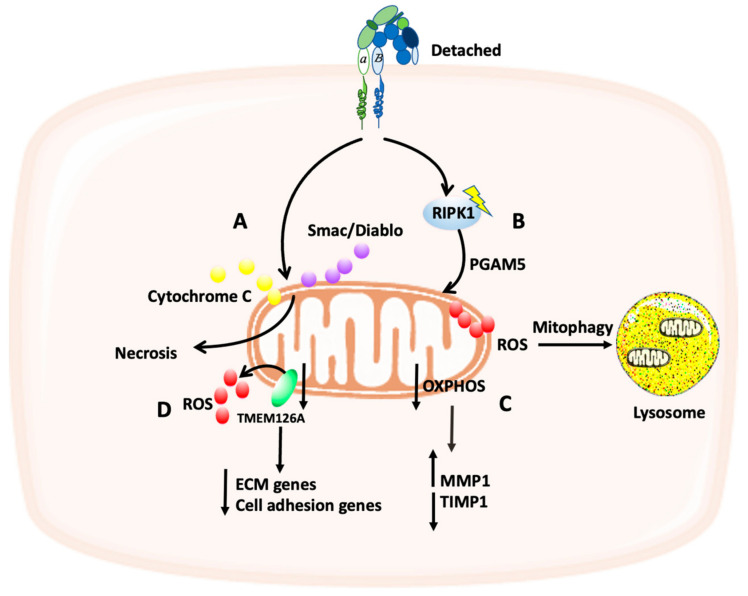
Schematic representation of the crosstalk between the ECM and mitochondria function. (**A**) ECM detachment can trigger the release of cytochrome C and Smac/Diablo from the mitochondria, resulting in necrosis. (**B**) ECM detachment can activate RIPK1 which in turn increases the production of ROS, leading to mitophagy. (**C**) OXPHOS deficiency results in increased MMP1 and degreased TIPM1 levels. (**D**) ROS production can be increased via the downregulation of TMEM126A, a mitochondria transmembrane receptor, resulting in reduced expression of ECM and cell adhesion genes.

## Data Availability

Data sharing not applicable, no new data were created or analysed in this study.

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
