# Peer review of "The Interplay between Cell-Extracellular Matrix Interaction and Mitochondria Dynamics in Cancer"

_cancers, 2022, doi:10.3390/cancers14061433_

Round 1

Reviewer 1 Report

Cancers review

Manuscript ID: cancers-1599914

Title: The interplay between cell-extracellular matrix interaction and mitochondria dynamics in cancer

Authors: Bian Yanes, Elena Rainero

Summary:

As indicated by the title, the authors have provided a review regarding the connections between cell- extracellular matrix (ECM) interactions and mitochondrial regulation in the context of cancer. Both authors are knowledgeable in this field and the corresponding author is an established expert and ideally placed to write a such a review. They have also published recent relevant manuscripts which are cited in the review. The review is timely, as it places recent data in the context of the literature, and relevant, as the mechanisms of mitochondrial regulation by cell-ECM interactions are an emerging interest in the field.

I am supportive of publication in the current format but have several suggestions that the authors may wish to consider including.

Comments:

  1. I would have liked more justification of the choice of the focus. Why in particular would cell-ECM interactions impact upon mitochondrial function? The authors have made a case for the relevance of both mitochondrial function and ECM in cancer but is it possible to better connect the two aspects? A brief paragraph in the introduction would suffice.
  2. Section 2 ‘Adhesion signalling modulates mitochondria dynamics and function’ would benefit from a more thorough description of both the complexity of ECM and integrin adhesion complexes. Perhaps this could be achieved with a brief introduction to the concepts of both the matrisome and adhesome? In particular, the subsequent description of the roles of specific adhesion complex proteins would be difficult to understand without this context.
  3. Also, for a reader that is not an expert in mitochondrial morphology and dynamics some basic information about the relevance of these aspects would help. You mention two proteins that regulate mitochondrial fission and fusion, but what is the relevance of fission/fusion to mitochondrial function? Perhaps the author could use the Cell Snapshot article on Mitochondrial Dynamics? (PMID: 21703455 DOI: 10.1016/j.cell.2011.06.018)
  4. Another aspect that could be used to strengthen the basis of the review would be to include information about links from cell-ECM adhesion to metabolism. The authors cite recent reviews (refs 11 and 23) that address this issue, and the corresponding author herself has recently published another in this field (ref 57). However, this information could be used to set this review, about links to mitochondria, in context of the wider functions of cell adhesion. A more formal short section covering this would be beneficial. The authors could include two relevant papers which are: Rabanal-Ruiz et al., mTORC1 activity is supported by spatial association with focal adhesions. J Cell Biol 3 May 2021; 220 (5): e202004010. doi: https://doi.org/10.1083/jcb.202004010 and Georgiadou et al., AMPK negatively regulates tensin-dependent integrin activity. J Cell Biol 3 April 2017; 216 (4): 1107–1121. doi: https://doi.org/10.1083/jcb.201609066
  5. In general paragraphs are too long and could be broken up to aid readability. Otherwise, the reader is faced by large chunks of intimidating text without a break.
  6. In both the Simple Summary (line 12) and Introduction (line 59) the authors suggest that the ECM only plays a tumour enhancing / promoting role. This is not the case. For example, in pancreatic cancer collagen deposition has been shown to inhibit cancer progression. Please adjust the language to stop possible misunderstanding by the reader.
  7. At times the language used is too informal. For example, in the Introduction (lines 35 and 39) the phrases ‘play a role in plenty of diseases’ and ‘mitochondria are divided into multiple structures to generate new ones’ could be rewritten. Please check throughout the manuscript for similar informal phrases.
  8. Line 62: Remove the strange symbol between ‘collagen’ and ‘matrix’.
  9. Line 72: You state: ‘the basement membrane (BM) manly composed of collagen IV, laminins, entactin and nidogen’. However entactin is nidogen and ‘manly’ should be ‘mainly’.
  10. Line 117: please rephrase ‘Parvin in this case disactivate the…’

Author Response

We would like to thank both reviewers for their in-depth analysis of our manuscript. They both raised important points which have all been taken into consideration when preparing this revised version of our review. We thank them for defining our review as “timely”, “relevant” and “identifying a relevant gap in the current knowledge”. We feel we have now addressed all the reviewers’ concerns, as listed below in this point-by-point response. All the changes in the text have been highlighted in red.

  1. I would have liked more justification of the choice of the focus. Why in particular would cell-ECM interactions impact upon mitochondrial function? The authors have made a case for the relevance of both mitochondrial function and ECM in cancer but is it possible to better connect the two aspects? A brief paragraph in the introduction would suffice: We have now added a paragraph in the introduction highlighting the important roles that the ECM plays in modulating cell metabolism, leading to the need for a better understanding of how the ECM might control mitochondria functions.
  2. Section 2 ‘Adhesion signalling modulates mitochondria dynamics and function’ would benefit from a more thorough description of both the complexity of ECM and integrin adhesion complexes. Perhaps this could be achieved with a brief introduction to the concepts of both the matrisome and adhesome? In particular, the subsequent description of the roles of specific adhesion complex proteins would be difficult to understand without this context: In the first 2 paragraphs in section 2, we have now added a more detailed description of the matrisome and the adhesome. We feel that this section is now providing all the required background information for readers to be able to fully understand the following sections
  3. Also, for a reader that is not an expert in mitochondrial morphology and dynamics some basic information about the relevance of these aspects would help. You mention two proteins that regulate mitochondrial fission and fusion, but what is the relevance of fission/fusion to mitochondrial function? Perhaps the author could use the Cell Snapshot article on Mitochondrial Dynamics? (PMID: 21703455 DOI: 10.1016/j.cell.2011.06.018): The introduction now includes 2 additional paragraph describing how fission and fusion impact on mitochondria function, also taking advantage of the suggested reference, which has now been included
  4. Another aspect that could be used to strengthen the basis of the review would be to include information about links from cell-ECM adhesion to metabolism. The authors cite recent reviews (refs 11 and 23) that address this issue, and the corresponding author herself has recently published another in this field (ref 57). However, this information could be used to set this review, about links to mitochondria, in context of the wider functions of cell adhesion. A more formal short section covering this would be beneficial. The authors could include two relevant papers which are:
    1. Rabanal-Ruiz et al., mTORC1 activity is supported by spatial association with focal adhesions. J Cell Biol 3 May 2021; 220 (5): e202004010. doi: https://doi.org/10.1083/jcb.202004010
    2. Georgiadou et al., AMPK negatively regulates tensin-dependent integrin activity. J Cell Biol 3 April 2017; 216 (4): 1107–1121. doi: https://doi.org/10.1083/jcb.201609066

               We have now added a new section on how cell-ECM interaction impinges on nutrient signalling, by including the suggested references.

  1. In general paragraphs are too long and could ne broken up to aid readability. Otherwise, the reader is faced by large chunks of intimidating text without a break. The paragraphs have now been broken up as suggested.
  2. In both the Simple Summary (line 12) and Introduction (line 59) the authors suggest that the ECM only plays a tumour enhancing / promoting role. This is not the case. For example, in pancreatic cancer collagen deposition has been shown to inhibit cancer progression. Please adjust the language to stop possible misunderstanding by the reader. We have now highlighted more clearly that the ECM can have both a tumour suppressive and a tumour promoting role.
  3. At times the language used is too informal. For example, in the Introduction (lines 35 and 39) the phrases ‘play a role in plenty of diseases’ and ‘mitochondria are divided into multiple structures to generate new ones’ could be rewritten. Please check throughout the manuscript for similar informal phrases. This instances have now been removed.
  4. Line 62: Remove the strange symbol between ‘collagen’ and ‘matrix’. This was a formatting issue and has now been removed.
  5. Line 72: You state: ‘the basement membrane (BM) manly composed of collagen IV, laminins, entactin and nidogen’. However, entactin is nidogen and ‘manly’ should be ‘mainly’. We apologise for this oversight, these have now been corrected .
  6. Line 117: please rephrase ‘Parvin in this case disactivate the… The sentence has now been rephrased.

Reviewer 2 Report

This review addresses an interesting topic, and the authors identify a relevant gap in the current knowledge. The main aim is to summarise the link between the extracellular matrix and mitochondria (function and dynamics) in the context of cancer, describing studies that report or potentially suggest the importance of this interplay. However, many points need to be addressed to consider the manuscript suitable for publication, mainly the connection between topics and concepts needs to be improved. Also, the main title and the subtitles do not adequately reflect the subjects presented and discussed in the abstract or in the main text (the attention is more on mitochondrial function and not only dynamics).

Major issues:

  1. Chapter 1: needs to be extended and more focussed to cover the main themes that will be discussed throughout the review as well as avoid those that are outside its scope. I suggest including a description of the mitochondrial changes observed in cancer. Moreover, topics must be only introduced, and the level of details shown in lines 45-51 or lines 59-62 are not required.
  2. Chapter 2: ROS paragraph (123-133) needs more introduction because it appears out of context, and it is not clear the link with the rest of the chapter.
  3. Chapter 3: most of the chapter is based on references not addressing cancer. This is fine to explain the mechanistic processes driving this connection. However, given the nature of the review, it would be good to compare it with reality in cancer. If that is not possible for lack of studies, that should be mentioned as well. Moreover, the connection between ECM remodelling and actin changes should be further explored in the context of cancer (line 167).
  4. Chapters 4, 5: need considerable work, they lack connection with ECM and they are hard to read. The first half of chapter 4 discusses the role of mitochondrial dynamics in cell migration, but no mention of ECM and it poorly connects with the rest of the paragraph. In the second half, interesting ideas are presented, however, again these need to be linked more coherently. Finally, the statement in line 216 is not accurate, as there are existing references explaining mechanisms of mitochondrial trafficking to the leading edges as well as others providing information on Drp1-actin interactions. For example: Furnish M, Cancer Rep (Hoboken). 2020 Feb;3(1):e1157. And Moore AS, Curr Opin Physiol. 2018 Jun;3:94-100.
  5. Chapter 6: The conclusion also requires rewriting. The first half (until 365) is completely focused on cancer metabolism and some sentences are too similar to the introduction. The last section (lines 370-378) is not related to the aims of the review.

Minor comments:

  • Line 38: mitochondrial dynamics include not just fission and fusion, and processes like mitochondrial intracellular location and trafficking, as well as mitophagy need to be added here, especially considering that are repeatedly described in the review.
  • Line 190: there is a mention of ECM stiffness, although the importance of this feature in cancer is not explained until the middle of chapter 4 (line 239).
  • Lines 344-346: does not seem to be relevant for this review.
  • Figures: Adding letters to refer to different parts of the figure would make it easier for the reader.
  • Line 33-34: seems to be redundant talking about calcium (which is not analysed in the main text)
  • Line 336: in accordance with the concept of a possible bidirectional interaction between mitochondria and ECM it is also relevant to mention that the same group who studied Col6 KO mice (ref 47 and 48) demonstrated that acting directly on the mitochondria it is possible to revert the dystrophic phenotype and possibly change the ECM (Palma et al. Human Molecular Genetics, Volume 18, Issue 11, 1 June 2009, Pages 2024–2031).
  • I would suggest adding a comment on the following article on Nature Cell Biology. I know that this paper just came out this week, but it would be very relevant for the review and a missed opportunity considering the review will be published afterward. Romani, P., Nirchio, N., Arboit, M. et al. Mitochondrial fission links ECM mechanotransduction to metabolic redox homeostasis and metastatic chemotherapy resistance. Nat Cell Biol 24, 168–180 (2022). 

Author Response

We would like to thank both reviewers for their in-depth analysis of our manuscript. They both raised important points which have all been taken into consideration when preparing this revised version of our review. We thank them for defining our review as “timely”, “relevant” and “identifying a relevant gap in the current knowledge”. We feel we have now addressed all the reviewers’ concerns, as listed below in this point-by-point response. All the changes in the text have been highlighted in red.

Chapter 1: needs to be extended and more focussed to cover the main themes that will be discussed throughout the review as well as avoid those that are outside its scope. I suggest including a description of the mitochondrial changes observed in cancer. Moreover, topics must be only introduced, and the level of details shown in lines 45-51 or lines 59-62 are not required. The introduction has been extended and more information about mitochondria dynamics and its role in cancer progression were added. In addition, we are also providing a more detailed introduction on the ECM and cell-ECM interaction

Chapter 2: ROS paragraph (123-133) needs more introduction because it appears out of context, and it is not clear the link with the rest of the chapter The introduction about ROS has been amended and the organization of this paragraph has been changed, so that it is now more clearly linked with the rest of the section.

Chapter 3: most of the chapter is based on references not addressing cancer. This is fine to explain the mechanistic processes driving this connection. However, given the nature of the review, it would be good to compare it with reality in cancer. If that is not possible for lack of studies, that should be mentioned as well. Moreover, the connection between ECM remodelling and actin changes should be further explored in the context of cancer (line 167). We apologise for the confusion created here. The studies referred to in this section where actually performed in cancer cells (Korobova et al. and Manor et al. were with U2OS which are osteosarcoma cells. Moor et al and Li et al 2015 studies were done using Hela cells and Li et al 2018 was done using MDA-MB-231 cells which are breast cancer cells). We have now clearly stated this in the text. In addition, we added a paragraph highlighting the importance of ECM-driven changes in the actin cytoskeleton associated with cancer.

Chapters 4, 5: need considerable work, they lack connection with ECM and they are hard to read. The first half of chapter 4 discusses the role of mitochondrial dynamics in cell migration, but no mention of ECM and it poorly connects with the rest of the paragraph. In the second half, interesting ideas are presented, however, again these need to be linked more coherently. Finally, the statement in line 216 is not accurate, as there are existing references explaining mechanisms of mitochondrial trafficking to the leading edges as well as others providing information on Drp1-actin interactions. For example: Furnish M, Cancer Rep (Hoboken). 2020 Feb;3(1):e1157. And Moore AS, Curr Opin Physiol. 2018 Jun;3:94-100. We have now more explicitly linked section 4 and 5 with the ECM, highlighting how this plays a fundamental role in controlling cell migration. To help with the readability of this section, we have broken down section 4 in 2 subsections. The first section discusses the role of mitochondria dynamics in cancer cells migration, focusing only on literature assessing this process in the presence of ECM. The second section specifically focus on the role of ECM stiffness in controlling mitochondria functions. The statement in line 216 was referring to the fact that, in the specific context of the paper, the molecular mechanisms through which mitochondria trafficking was regulated had not been addressed. We apologised that this caused a misunderstanding and we have now remove that statement.  Chapter 4 has been divided into two sections to make it easier. The first section discusses the role of mitochondria dynamics in cancer cells migration. The studies mentioned in this section performed the experiments in the presence of ECM. However, they haven’t mentioned if the ECM can affect the results and that’s why this was followed by the second section of the chapter showing that the ECM stiffness can affect mitochondria. 

Chapter 6: The conclusion also requires rewriting. The first half (until 365) is completely focused on cancer metabolism and some sentences are too similar to the introduction. The last section (lines 370-378) is not related to the aims of the review. The conclusion section has been completely rewritten, and has now a stronger focus on the content of this review. We still believe in linking this with the broad cancer metabolism field, and this is why we kept some of the cancer metabolism points raised before.

Minor comments:

  • Line 38: mitochondrial dynamics include not just fission and fusion, and processes like mitochondrial intracellular location and trafficking, as well as mitophagy need to be added here, especially considering that are repeatedly described in the review. We have now included more information about mitophagy in the introduction.
  • Line 190: there is a mention of ECM stiffness, although the importance of this feature in cancer is not explained until the middle of chapter 4 (line 239). More information about ECM stiffness has now been added at the beginning of section 3
  • Lines 344-346: does not seem to be relevant for this review. We strongly believe that this paragraph is relevant and important for this review. The point we are making here is that there is a reciprocal interaction between ECM and mitochondria, whereby the ECM can control mitochondria function – as highlighted in this review – but also mitochondria dysfunctions can impact on ECM remodelling. This section in particular focuses on the impact of OXPHOS and mitochondria depolarisation in controlling ECM dynamics and cell-ECM adhesion. We have now stated this point more clearly in the text.
  • Figures: Adding letters to refer to different parts of the figure would make it easier for the reader. We fully agree with this recommendation, the addition of letters allows better links between the text and the figures. These have now been included in all figures and the figure legends have been modified accordingly.
  • Line 33-34: seems to be redundant talking about calcium (which is not analysed in the main text) We have now added more details in the text explaining the role of Ca++ in this process.
  • Line 336: in accordance with the concept of a possible bidirectional interaction between mitochondria and ECM it is also relevant to mention that the same group who studied Col6 KO mice (ref 47 and 48) demonstrated that acting directly on the mitochondria it is possible to revert the dystrophic phenotype and possibly change the ECM (Palma et al. Human Molecular Genetics, Volume 18, Issue 11, 1 June 2009, Pages 2024–2031). We kindly thank the reviewer for suggesting this reference, which strengthens our concept. We have now included this in the text.
  • I would suggest adding a comment on the following article on Nature Cell Biology. I know that this paper just came out this week, but it would be very relevant for the review and a missed opportunity considering the review will be published afterward. Romani, P., Nirchio, N., Arboit, M. et al. Mitochondrial fission links ECM mechanotransduction to metabolic redox homeostasis and metastatic chemotherapy resistance. Nat Cell Biol 24, 168–180 (2022). We fully agree with the importance of adding a paragraph on this new study, which is extremely relevant for this review. This has now been discussed in the text.

Round 2

Reviewer 2 Report

As stated before, the review topic is very relevant and timely. The Authors have considerably improved the article, which is easy to follow, with a precise focus and well-connected paragraphs. All the comments have been adequately addressed.